# Real-space imaging of acoustic plasmons in large-area graphene grown by chemical vapor deposition

Sergey G. Menabde [1,7], In-Ho Lee[2,6,7], Sanghyub Lee[3,4], Heonhak Ha[1], Jacob T. Heiden [1], Daehan Yoo[2], Teun-Teun Kim [3,5], Tony Low[2], Young Hee Lee [3,4✉], Sang-Hyun Oh [2✉] & Min Seok Jang [1✉]

An acoustic plasmon mode in a graphene-dielectric-metal structure has recently been spotlighted as a superior platform for strong light-matter interaction. It originates from the coupling of graphene plasmon with its mirror image and exhibits the largest field confinement in the limit of a sub-nm-thick dielectric. Although recently detected in the far-field regime, optical near-fields of this mode are yet to be observed and characterized. Here, we demonstrate a direct optical probing of the plasmonic fields reflected by the edges of graphene via near-field scattering microscope, revealing a relatively small propagation loss of the mid-infrared acoustic plasmons in our devices that allows for their real-space mapping at ambient conditions even with unprotected, large-area graphene grown by chemical vapor deposition. We show an acoustic plasmon mode that is twice as confined and has 1.4 times higher figure of merit in terms of the normalized propagation length compared to the graphene surface plasmon under similar conditions. We also investigate the behavior of the acoustic graphene plasmons in a periodic array of gold nanoribbons. Our results highlight the promise of acoustic plasmons for graphene-based optoelectronics and sensing applications.

[1] School of Electrical Engineering, Korea Advanced Institute of Science and Technology (KAIST), Daejeon, Korea. [2] Department of Electrical and Computer Engineering, University of Minnesota, Minneapolis, USA. [3] Center for Integrated Nanostructure Physics (CINAP), Institute for Basic Science (IBS), Suwon, Korea. [4] Department of Energy Science, Sungkyunkwan University, Suwon, Korea. [5] Department of Physics, University of Ulsan, Ulsan, Korea. [6] Present address: Center for Opto-Electronic Materials and Devices, Korea Institute of Science and Technology, Seoul, Korea. [7] These authors contributed equally: Sergey G. Menabde, In-Ho Lee. ✉email: leeyoung@skku.edu; sang@umn.edu; jang.minseok@kaist.ac.kr

An acoustic plasmon mode supported by a system of two graphene sheets[1], or by a single graphene sheet over a metal gate[2], was experimentally detected recently[3], and its unprecedented field confinement can benefit applications in the technologically important mid-infrared (MIR) and THz regimes[4–8]. This mode is supported by a structure comprising of a metal, a dielectric spacer, and a graphene layer, where the image charges in the metal effectively "mirror" the charge density oscillations in the doped graphene layer. The acoustic graphene plasmon (AGP) supported by the structure is mostly confined in the dielectric spacer and does not experience cutoff as the spacer thickness decreases[8], thus resembling the fundamental plasmonic mode of a narrow metal gap. The AGP excited at the important MIR frequencies[9,10] does not exhibit significant loss and is detectable even when the spacer is reduced to a single atomic layer of hexagonal boron nitride[5]. Inside such a narrow dielectric spacer, the AGP wavevector can be about two orders of magnitude larger than that of free space light, which grants access to quantum and non-local phenomena in graphene[5,11,12], and allows for the AGP localization in nanostructures[13] with a stunning mode volume confinement factor[4] of ~$10^{10}$. This ultimate capability to compress MIR light outperforms that of other polaritonic species in van der Waals materials[9], including graphene surface plasmon (GSP)[14,15], and is similar to the case of image phonon-polaritons in boron nitride[16]. For that reason, AGP is promising for applications that require strong light–matter interaction such as molecular sensing[6,17–20], polaritonic dispersion engineering in van der Waals crystals[21,22], and dynamic light manipulation by graphene-based active metasurfaces[23–26].

The key advantage of the AGP is its confinement within the dielectric spacer, in contrast to the GSP bound to the graphene layer. Therefore, ohmic losses in graphene are expected to hinder the AGP propagation to a lesser extent compared to GSP. On the other hand, the larger AGP wavevector requires an intermediary structure to alleviate the momentum mismatch under the far-field excitation. So far, structures containing an array of metallic elements have been used to couple far-field radiation to the AGP mode[4–6]. However, the near-field optical probing of AGP, which provides detailed microscopic information about the mode, is yet to be demonstrated.

In this work, we employ a scattering-type scanning near-field optical microscope (s-SNOM) based on an atomic force microscope (AFM) for real-space mapping and analysis of MIR AGP. We directly measure the AGP dispersion, evaluate the mode propagation loss, and investigate its behavior in a periodic structure designed for the far-field coupling. Most importantly, our results reveal a relatively low propagation loss of infrared AGP even when unprotected, CVD-grown, large-area graphene is used at ambient conditions, suggesting a practical route to construct large-area graphene-based optoelectronic devices.

## Results

**Near-field coupling to AGP.** Although the AGP fields are mainly confined inside the dielectric spacer, their evanescent components have non-zero amplitudes above the graphene layer (Fig. 1a). The vertical component of the mode's electric field, $E_z$, penetrates into the free space above the structure, and hence, can couple to the AFM tip of the s-SNOM effectively acting as a $z$-oriented electric dipole[27–30] (Fig. 1b). We define the $E_z$ penetration depth above graphene, $D_e$, as corresponding to the 1/e attenuation of the field amplitude. By solving Maxwell's equations in a multilayer configuration[31], it is possible to find the unique solution for the AGP eigenmode supported by the structure of interest (see Supplementary Information section S-1). Then, $D_e = 1/\text{Im}\{k_z\}$, where $k_z$

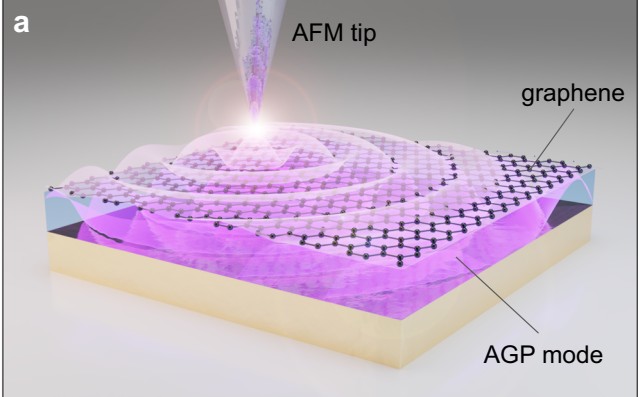

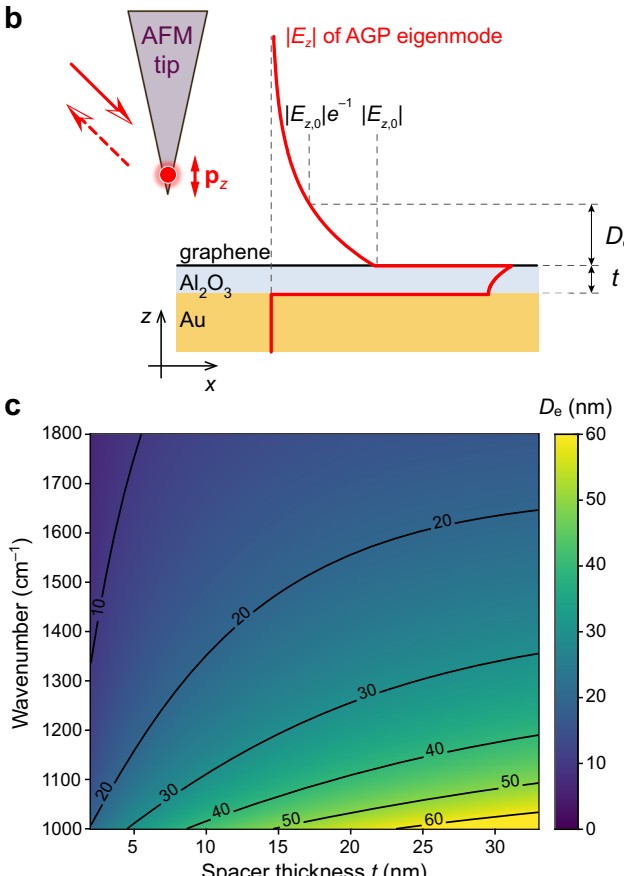

**Fig. 1 Plasmon coupling to the AFM tip. a** The AFM tip couples to the AGP inside the layered structure via the evanescent field above graphene. **b** The exponentially decaying $z$-component of the AGP electric field $E_z$ penetrates into the free space above graphene by distance $D_e$ and couples to the AFM tip that acts as a $z$-oriented electric dipole $\mathbf{p}_z$. **c** Penetration depth of $|E_z|$ as a function of excitation frequency and spacer thickness calculated for graphene $E_F = -0.5$ eV.

is the $z$-component of the AGP wavevector in the medium above graphene. At a given graphene Fermi level, $E_F$, the wavevector depends on both the excitation frequency $\omega$ and the spacer thickness $t$, and so does the $D_e$. At the same time, the amplitude of the scattered near-field signal from the AFM tip is proportional to $|E_z|$ at the position of the tip. Therefore, the performance of the s-SNOM method is expected to vary significantly depending on the experimental conditions.

In order to estimate the optimal experimental conditions for the near-field AGP probing by s-SNOM, it is instructive to calculate

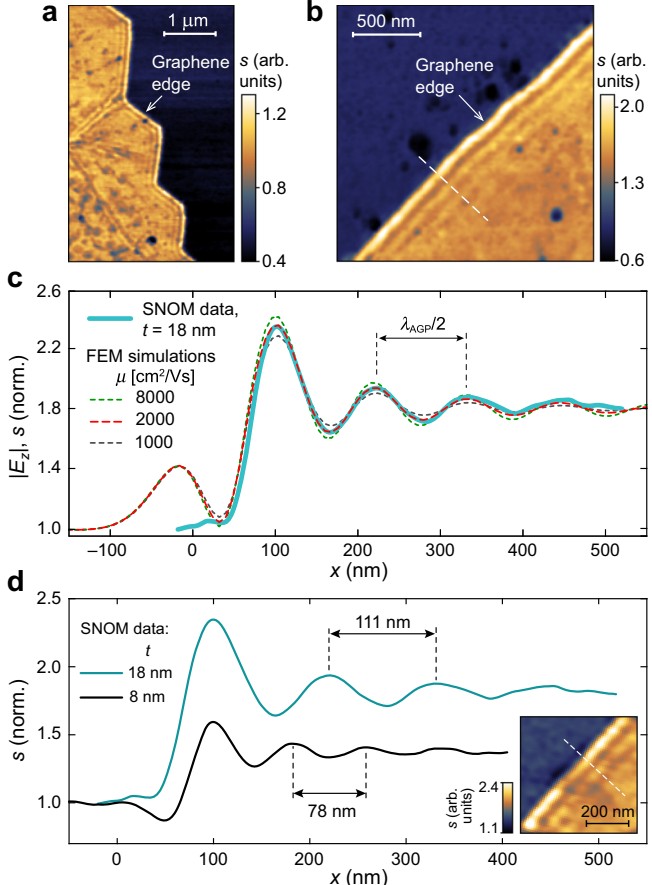

**Fig. 2 Near-field mapping of the AGP interference fringes at the graphene edge. a** Distribution of the near-field signal intensity $s(x,y)$ over the doped graphene near its edge, where the AGP interference fringes are visible. **b** High resolution $s(x,y)$ scan of the graphene edge with AGP interference fringes. **c** Near-field signal intensity (blue solid) measured across the edge shown in (**b**) along the white dashed line (averaged over a ten-pixels-wide line), and calculated $|E_z|$ at different carrier mobility in graphene (dashed) with $E_F = -0.51$ eV. **d** AGP fringes across the graphene edge in samples with spacer thickness $t = 18$ nm (blue) and 8 nm (black; fitted $E_F = -0.49$ eV). Inset: $s(x,y)$ over the sample with $t = 8$ nm. All data is at $\omega = 1150$ cm$^{-1}$.

$D_e(\omega,t)$ for the structure of interest. Figure 1c shows $D_e(\omega,t)$ calculated for the range of MIR frequencies and spacer thicknesses, assuming $E_F = -0.5$ eV. Considering that the average tip height above the sample is 40–80 nm, which is approximately equal to the tapping amplitude of the tip[29,32], the most favorable experimental conditions are expected in the frequency window of 1000–1300 cm$^{-1}$ (where Al$_2$O$_3$ absorption stays sufficiently low), while $t > 10$ nm.

Substrates with gold and Al$_2$O$_3$ films were fabricated using the template-stripping method[33,34], allowing a sub-nm roughness of substrate surface even when the gold film is patterned[6] (see Supplementary Information section S-2). The large-area mono-crystalline graphene[35] was grown by chemical vapor deposition (CVD), wet-transferred on top of the alumina-coated gold substrates, and chemically doped (see Methods section for details).

**Dispersion and loss analysis.** Near-field imaging of doped graphene on the Au/Al$_2$O$_3$ substrate reveals an abundance of μm-long edges of monocrystalline areas with AGP interference fringes

formed due to its reflection from the graphene termination (Fig. 2a). Figure 2b demonstrates a close-up scan of such an edge for a sample with $t = 18$ nm at $\omega = 1150$ cm$^{-1}$. The near-field signal intensity $s(x,y)$ is proportional to the amplitude of the z-component of the electric field under the tip[30], $s \propto |E_z|$, thus it can be numerically calculated by full-wave simulations in a quasi-static approximation[29,36] (see Supplementary Information section S-3). As shown in Fig. 2c, the full-wave simulations by finite element method (FEM) with AFM tip modeled as a point dipole source provide a perfect fit to the interference pattern, where numerical and experimental data normalized by that from the graphene-free area. The fitted value of the optical conductivity of graphene (given by the random phase approximation in local limit[37–39]) corresponds to $E_F = -0.51$ eV and carrier mobility $\mu = 2000$ cm$^2$/V s, consistent with a high-quality CVD graphene[7]. The roughness-mediated scattering of AGP is neglected in the full-wave simulations. Additional simulations support this assumption, considering the measured root-mean-square (RMS) roughness of 0.5 nm at gold and alumina surfaces (Supplementary Information section S-2).

The AGP interference fringes allow for the direct measurement of the plasmonic wavelength[30,40] $\lambda_{AGP} \approx 222$ nm (Fig. 2c, d); note that this is not the case for graphene patches of finite size where multiple reflections contribute to near-field signal[30]. Additionally, we observe AGP interference in a sample with $t = 8$ nm (inset in Fig. 2d). As expected, the AGP mode in the thinner 8 nm spacer is more tightly confined in the gap, hence the shorter $\lambda_{AGP} \approx 156$ nm and the weaker amplitude of the near-field signal above graphene, despite the similar doping level $E_F = -0.49$ eV. We note that the probed structure with $t \ll \lambda_{AGP}$ does not support propagating GSP due to the proximity of graphene to the gold layer. For the MIR frequencies of interest and $E_F \approx -0.5$ eV, it has been analytically demonstrated[8] that the AGP dispersion approaches that of the GSP when $t \gtrsim 100$ nm.

By plugging the recovered graphene conductivity into the semi-analytic eigenmode solver, the parameters of the detected AGP can be obtained. For the sample with $t = 18$ nm (8 nm), the effective index of the AGP is $q_{AGP} = k_{AGP}/k_0 = 39.06 + 2.92i$ (55.96 + 4.17i), where $k_{AGP}$ is the propagation constant of the AGP, and $k_0 = 2\pi/\lambda_0$ is that of free space light. To quantify the dissipation of plasmonic modes, we use the figure of merit (FOM) defined as the ratio of the plasmon propagation length $L_p$ to its wavelength $\lambda_p$: $L_p/\lambda_p = l_p = \text{Re}\{k_p\}/(2\pi\text{Im}\{k_p\})$, where $l_p$ is the normalized plasmon propagation length in optical cycles. It can be noted that the FOM of AGP $l_{AGP} = 2.12$ (in both samples) is almost twice larger than the reported value for MIR GSP in exfoliated graphene on an SiO$_2$ substrate[41] $l_{GSP} = 1.18$. An analytical solution for GSP in the same graphene on a thick Al$_2$O$_3$ substrate provides $q_{GSP} = 23.41 + 2.47i$ (24.43 + 2.62i), and thus, an expected $l_{GSP} \approx 1.5$ in both cases. Therefore, our near-field measurements indicate that, while the observed AGP is ×1.7 (×2.3) times more compressed in terms of the wavenumber, its FOM is 1.4 times higher than that of the GSP in the same graphene sheet. The better FOM of AGP has been predicted[8] due to the larger compression factor of the plasmonic wavelength compared to the decrease in the propagation distance. However, dispersion analysis also indicates that the AGP is in general less sensitive to the loss in graphene if considered in local limit[42] (see Supplementary Information section S-1). Nonetheless, the experimental observation of the low-loss AGP with unprotected CVD graphene at ambient conditions is encouraging for the development of large-area polaritonic devices operating in the MIR[9].

The AGP dispersion can be directly measured from the near-field images at different frequencies. The AGP dispersion measured in a sample with $t = 21$ nm (circles) is shown in Fig. 3a, along with the fitted analytical dispersion for $E_F = -0.46$ eV. The near-field data are obtained from a series of measurements over

the same sample area, which makes it possible to compare the spectral dependency of the near-field contrast[29,32] $\eta(x,y) = (s(x,y)/s_{\text{ref}})e^{i(\varphi(x,y)-\varphi_{\text{ref}})}$, where $s_{\text{ref}}$ and $\varphi_{\text{ref}}$ are the amplitude and phase of the near-field signal over the graphene-free area, respectively. Figure 3b demonstrates mapping of $|\eta(x,y)| = s(x,y)/s_{\text{ref}}$ and corresponding AGP interference fringes at different frequencies, indicating the effective mode index increasing from 34 at 1080 cm$^{-1}$ to 46 at 1260 cm$^{-1}$. Furthermore, the spectral dependency of $|\eta|$ above graphene (averaged across the area far from the edge) generally follows the calculated value of $D_{\text{e}}$ (Fig. 3c), in agreement with the tighter mode confinement inside the spacer. According to Fig. 3c, $|\eta|$ approaches unity when $D_{\text{e}} \approx 25$ nm. Therefore, based on the calculations for $D_{\text{e}}$ shown in Fig. 1c, we predict that the s-SNOM technique would be feasible for AGP probing even when the spacer thickness is reduced down to a few nanometers if $\omega$ is sufficiently low.

**AGPs in periodic structures.** Due to the significant momentum mismatch, the efficient AGP coupling to a far-field requires a mediator—an array of metallic elements (e.g. gold nanoribbons), which can provide coupling efficiency exceeding 90% when combined with an optical cavity[6]. We fabricated AGP resonators similar to those used in Ref. [6], where gold nanoribbons are embedded in the alumina layer (Fig. 4a). Due to the periodicity and finite width of the nanoribbons, their interaction with AGP may produce non-trivial near-field patterns depending on the ratio between the AGP wavelength, array period $P$, and ribbon width $w$. We investigate samples with different $w$, while the gap size is 30 nm and $t = 18$ nm in all devices; the effect of the underlying cavity is not considered in this study.

The near-field signal from a non-uniform structure bears information from multiple scattering sources. Therefore, the AGP propagation loss cannot be extracted from the interference fringes. At the same time, $\lambda_{\text{AGP}}$ does not depend on the geometry of the structure, thus $E_{\text{F}}$ in graphene can be fitted using the AGP dispersion obtained from the near-field imaging (see Supplementary Information section S-4). Figure 4b demonstrates the spatial distribution of $|\eta|$ at different frequencies, measured over the same area of graphene deposited on alumina with embedded gold nanoribbons ($w = 240$ nm). Dispersion fit to $\lambda_{\text{AGP}}$ at different frequencies (Fig. 4c) provides $E_{\text{F}} = -0.58$ eV, while the $q_{\text{AGP}}$ increases from 32 at 1080 cm$^{-1}$ to 44 at 1280 cm$^{-1}$. The correlation between $|\eta|$ and $D_{\text{e}}$ (Fig. 4d) is very similar to that observed in the uniform structure, indicating a stronger AGP confinement at higher frequencies, while $|\eta|$ approaches unity at $D_{\text{e}} \approx 30$ nm.

In our experiments, the plane of incidence of the TM-polarized excitation beam is always orthogonal to the nanoribbons in order to maximize the scattering at the metal edges (as indicated by the red arrow in the first panel of Fig. 4b). While the AFM tip is able to excite AGP with an arbitrary direction and magnitude of the wavevector, the excitation beam is expected to couple only to the mode propagating in the periodic structure across the ribbons, with maximum coupling efficiency reached at the phase-matching condition between the AGP and the array[6] $k_{\text{AGP}} = 2\pi/P$. One might expect that the near-field signal can be enhanced at the phase-matching condition. However, the near-field contrast over the nanoribbons (measured far from the graphene edge; Fig. 4d) does not show any noticeable feature around the phase-matching frequency of 1105 cm$^{-1}$. To understand this and gain an insight into the near-field excitation of AGP in the array, we calculate the dispersion of AGP propagating in a periodic array of nanoribbons.

The AGP dispersion in the $x$-direction (across the nanoribbons) is calculated using a simple model of a planar AGP waveguide with an infinite array of nanogaps, treated as partially reflective "mirrors" with complex transmission and reflection coefficients. Then, the dispersion solution is reduced to the eigenvalue problem for a lossy Bloch state in a 1D periodic medium[43]. The periodicity of the structure does not lead to the opening of a bandgap or flattening of bands at the center or edges of the Brillouin zone, possibly due to the lossy nature of plasmonic modes (see Supplementary Information section S-5). As a result, the density of optical states has similar value throughout the measured spectral range. Therefore, our measurements of the near-field contrast (Fig. 4) do not show stronger near-field signal at the frequency of the phase matching.

We proceed with analysis of the several instances of near-field images. We first analyze the case of phase matching between the array and the AGP. Figure 5a shows the spatial distribution of near-field signal intensity $s(x,y)$ and phase $\varphi(x,y)$ obtained at $\omega = 1200$ cm$^{-1}$ in the sample with $P = 230$ nm and $E_{\text{F}} = -0.62$ eV; $\lambda_{\text{AGP}} \approx 228$ nm, so that $\beta_{\text{AGP}} = 2\pi/P$. The measured and the numerically calculated profiles of $s \propto |E_z|$ and $\varphi \propto \arg\{E_z\}$ across the nanoribbons (Fig. 5b) both show a periodic variation with the period $P$ equal to the plasmonic wavelength, as demonstrated by the identical red scale bars on the left panel of Fig. 5a. Furthermore, the electric field amplitude (phase) has its maxima (minima) over the center of the nanoribbons, while the minima (maxima) are aligned with the nanogaps (Fig. 5b). When the AGP momentum starts to exceed that of the array, the near-field patterns of both amplitude and phase drastically change, as demonstrated in Fig. 5c, d for the sample with $P = 260$ nm, $\omega = 1150$ cm$^{-1}$, graphene $E_{\text{F}} = -0.52$ eV, and $\lambda_{\text{AGP}} \approx 225$ nm, so that $k_{\text{AGP}} = 1.15 \times 2\pi/P$. Now, the field maxima are recorded over the gaps, while the minima are at the centers of the nanoribbons. The difference in the near-field distribution can be attributed to the excitation of different eigenmodes in the periodic structure (see Supplementary Section S-6 for detailed analysis of eigenmodes in the structure).

The near-field data in Fig. 5b, d is collected far from the graphene edge where the contribution of the edge-reflected AGP is minimized, which allows for employing a simplified model for 2D full-wave simulations using an infinitely long line dipole instead of a point dipole; without a far-field excitation source. Yet the simulation results show a qualitative agreement with the measurements. At higher frequencies, when the AGP wavelength is significantly smaller than $P$ and $w$, the near-field mapping reveals the AGP reflection and scattering at nanoribbon edges and nanogaps. Even then, the simple numerical model successfully renders the near-field distributions (see Supplementary Information section S-6).

## Discussion

In conclusion, we employ the near-field coupling for the excitation and direct optical probing of AGP at MIR frequencies. Since the AGP field is tightly confined in the dielectric region underneath graphene, direct near-field optical imaging of the AGP fields has been considered very challenging. Nevertheless, with a highly sensitive s-SNOM system and a high-quality CVD graphene sample, we are able to directly probe the relatively weaker evanescent tail of AGP above the graphene layer. Furthermore, near-field imaging reveals a lower propagation loss of MIR AGP compared to the reported earlier FOM of GSP in exfoliated graphene, even with unprotected CVD graphene at room temperature. The probed AGP mode is up to 2.3 times more confined than the GSP under similar conditions yet exhibits a 1.4 times

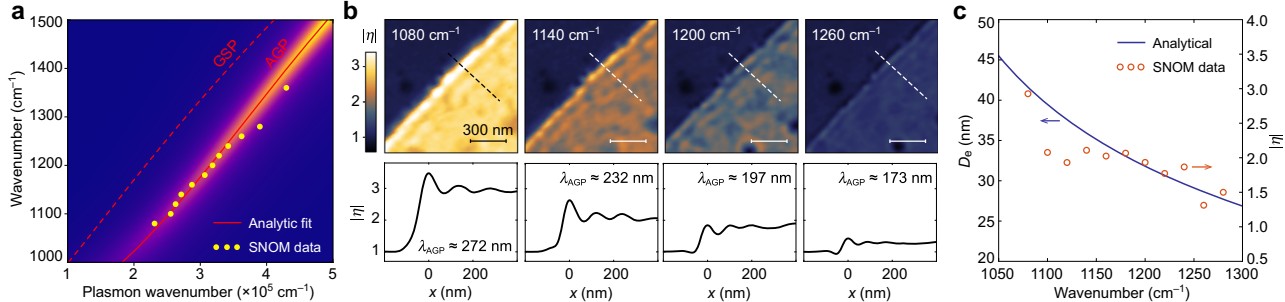

**Fig. 3 Plasmon dispersion. a** AGP dispersion obtained from the interference fringes in near-field at the sample with $t = 21$ nm (circles), and the analytically calculated dispersion for $E_F = -0.46$ eV: the exact solutions for AGP (red solid), GSP (red dashed; for thick alumina layer), and the imaginary part of the reflection coefficient (color map). **b** Top row: distribution of the near-field contrast $|\eta(x,y)|$ over the same graphene edge obtained at different excitation frequencies. Bottom row: corresponding $|\eta|$ profiles across the graphene edge, measured along the dashed line (average value for ten-pixel-wide lines). **c** Spectral dependency of the calculated $D_e$ (solid) and the measured $|\eta|$ above the sample shown in (**b**) (circles; averaged over the area far from the edge).

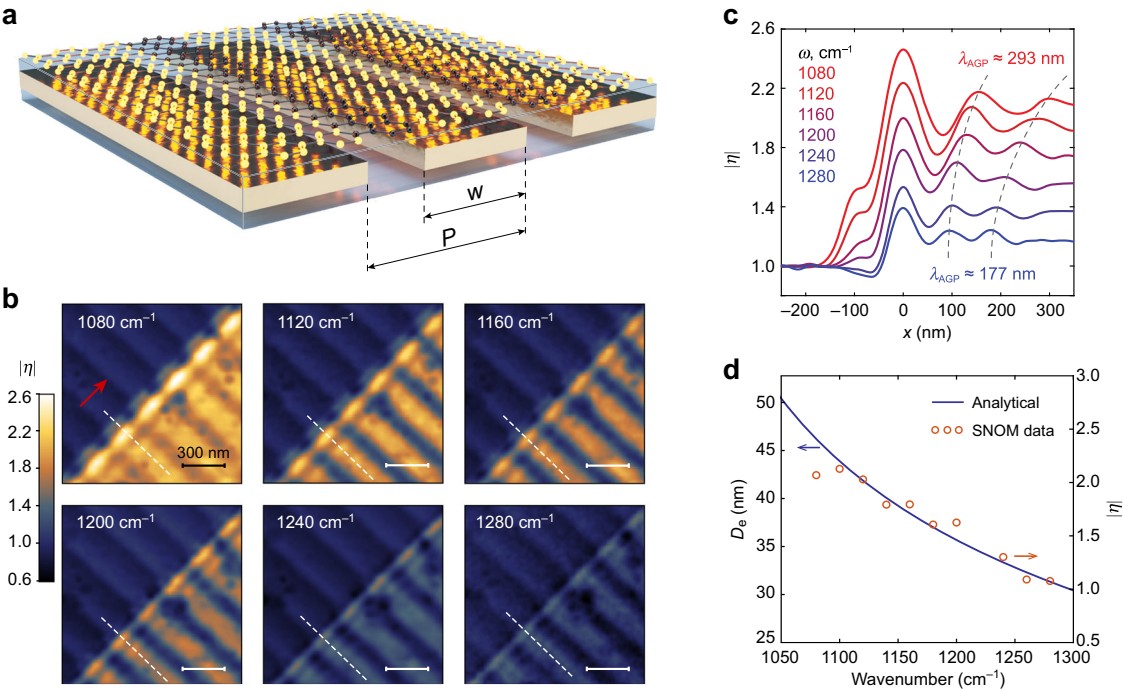

**Fig. 4 Near-field imaging of AGP over the periodic array of gold nanoribbons embedded in alumina. a** Schematics of the structure with gold nanoribbons of width $w$ arranged in an array with period $P$. **b** Near-field contrast $|\eta(x,y)|$ obtained at different frequencies over the same sample area; $P = 270$ nm, $w = 240$ nm, and $t = 18$ nm. Red arrow indicates the tip illumination direction. **c** Profile of $|\eta|$ across the graphene edge along the dashed lines in (**b**) showing smaller AGP wavelength and $|\eta|$ at higher excitation frequencies, indicating the stronger AGP confinement. **d** Spectral dependency of maximal $|\eta|$ above graphene measured far from the edge (circles) and calculated $D_e$ (solid) for the sample shown in (**b**).

larger FOM. These results highlight the promise of the AGP platform for investigating strong light–matter interactions and building graphene-based optoelectronic devices.

## Methods

**Device fabrication.** The gold/alumina layers on a Si substrate were prepared with template stripping as described in Ref. [6]. Large-area monocrystalline graphene was chemically grown on a single-crystal Cu foil. First, a commercial Cu foil (Nilaco Corporation, Japan) of 30 μm thickness was cut into ribbons and placed inside a CVD quartz tube, stretching between the hottest and the coldest zones inside the tube. Then, cycle annealing was introduced with a thermal gradient along the ribbons. The Cu foil was annealed at 1040 °C for two hours in an atmosphere of 40 sccm hydrogen and 1000 sccm argon gases. Then temperature was decreased to 700 °C during 30 min, and then increased up to 1040 °C during the same time. This process was repeated for four cycles in total, after which we opened the chamber to cool naturally.

For growing the high-quality graphene, we used low-concentration methane (0.1% in argon) in four stages: ramping, annealing, growth, and cooling. First, the temperature was increased up to 1060 °C during one hour and then kept stable for one hour for annealing, which is necessary for removing organic molecules and enlarging the Cu grain size. Then, we used a mix of three gases ($CH_4$, $H_2$, Ar) for graphene synthesis. The graphene flake size is controlled by growth conditions such as the ratio between $CH_4$ and $H_2$ concentrations, the total amount of $CH_4$, and the growth time. Here, we purged 5 sccm of $CH_4$, 30 sccm of $H_2$, and 1000 sccm of Ar for a full coverage of Cu by graphene. Then, Cu ribbons were cut and graphene was wet-transferred from Cu onto the prepared samples, and chemically doped by vapors of $HNO_3$ acid by placing the devices over the acid for 4 min at room temperature in a fume hood.

**Device characterization.** The near-field scans were obtained by commercial s-SNOM (Neaspec GmbH) coupled with a tunable quantum cascade laser (Daylight Solutions, MIRcat), which illuminates the Pt-coated AFM tip (Nano World,

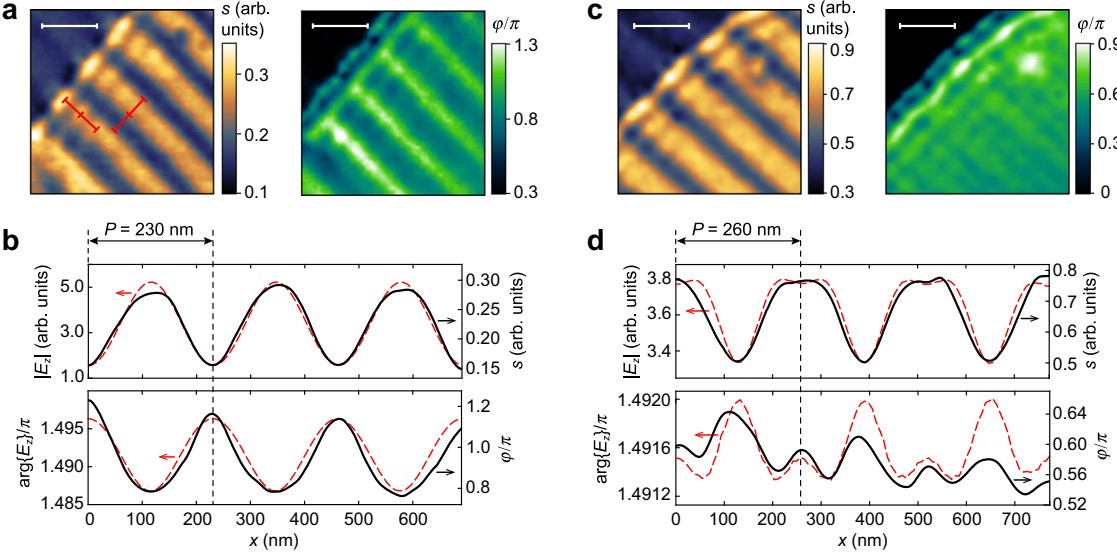

**Fig. 5 Near-field profile of AGP across the gold nanoribbons. a** Near-field signal amplitude $s(x,y)$ and phase $\varphi(x,y)$ at the sample with $P = 230$ nm ($t = 18$ nm, $E_F = -0.62$ eV, $\lambda_{AGP} \approx 228$ nm) at $\omega = 1200$ cm$^{-1}$ when the AGP momentum $k_{AGP}$ is similar to the array momentum $2\pi/P$ (indicated by the identical red scale bars of 230 nm). **b** Profiles of $s(x,y)$ (top panel) and $\varphi(x,y)$ (bottom panel) shown in (**a**), measured across the nanoribbons (black solid), and the numerically obtained (red dashed) by the full-wave FEM simulations $|E_z|$ (top panel) and arg$\{E_z\}$ (bottom panel). **c** Same as in (**a**), measured at the sample with $P = 260$ nm ($t = 18$ nm, $E_F = -0.52$ eV, $\lambda_{AGP} \approx 225$ nm) at $\omega = 1150$ cm$^{-1}$ when $k_{AGP} > 2\pi/P$. **d** Same as in (**b**), showing the irregular near-field profile attributed to the mixed signal form the several array modes. White scale bars are 300 nm.

ARROW-NCPt). The background-free interferometric signal[44], demodulated at third harmonic 3Ω (where Ω is the tapping frequency of the AFM tip), was used for near-field imaging. s-SNOM in AFM tapping mode was used to perform surface scans with 5 nm step and tip oscillation amplitude of ≈70 nm.

**Numerical simulations.** Commercial finite element method software (COMSOL Multiphysics) was used for full-wave simulations. In our 2D simulations, graphene is implemented as a thin film of finite thickness $\alpha = 0.2$ nm, having the effective relative dielectric permittivity $\varepsilon = \varepsilon_r + i\sigma/(\omega\varepsilon_0\alpha)$, where $\varepsilon_r$ is the background relative permittivity and $\sigma$ is the optical conductivity of graphene. Dielectric permittivity of gold was taken from Ref. [45], and that of thin film Al$_2$O$_3$ was taken from Ref. [46]. See Supplementary Information for details on full-wave simulations.

## Data availability
The data that support the findings of this study are available from the corresponding author upon reasonable request.

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

## Acknowledgements

This work was supported by the Samsung Research Funding & Incubation Center of Samsung Electronics under Project Number SRFC-IT1702-14. S.G.M. acknowledges support from the Young Researchers program of the National Research Foundation of Korea (NRF) funded by the Korean government (MSIT) (2019R1C1C1011131). I.-H.L., T.L., and S.-H.O. acknowledge support from the U.S. National Science Foundation (NSF ECCS 1809723). S.-H.O. further acknowledges support from the Samsung Global Research Outreach (GRO) Program and the Sanford P. Bordeau Chair in Electrical Engineering at the University of Minnesota. T.-T.K. acknowledges support from the Priority Research Centers Program through the NRF funded by the Ministry of Education (NRF-2019R1A6A1A11053838). S.L. and Y.H.L. acknowledge support from the Institute for Basic Science of Korea (IBS-R011-D1). Device fabrication was conducted in the Minnesota Nano Center, which is supported by the U.S. National Science Foundation through the National Nano Coordinated Infrastructure Network (NNCI) under Award Number ECCS-2025124.

## Author contributions

S.G.M., I.-H.L., S.-H.O., and M.S.J. conceived the idea. S.G.M. conducted the near-field measurements, analyzed the data, and wrote the manuscript. I.-H.L. and D.Y. fabricated the samples. S.L., T.-T.K., and Y.H.L. synthesized monocrystalline CVD graphene. H.H. and J.T.H. assisted in sample preparation and measurements. T.L. and M.S.J. analyzed the data and wrote the manuscript. Y.H.L., S.-H.O., and M.S.J. supervised the project.

## Competing interests

The authors declare no competing interests.
