## [Peer Review File · Nature Communications]

Reviewers' Comments:

Reviewer #1:

Remarks to the Author:

The paper presents the near-field study (by means of s-SNOM) of the screened ("acoustic") plasmon polaritons (AGPs) in CVD graphene. This is the first time the AGPs are visualized by purely optical means in CVD graphene. The manuscript is very well written and organized, as is typical for the research teams represented in the study. These results, undoubtedly, will be very interesting and useful for the large part of the community working on the optical properties of 2D materials. I strongly recommend this paper for publication in Nature Communications. However, before the paper is accepted, I would like to make a minor comment.

The authors have experimentally found that the propagation length in terms of the wavelength of the AGPs (the authors call this quantity as the damping rate) is larger than that of the conventional plasmon polaritons in graphene (GPs). Although they find the same increase of the damping rate in the theory (assuming the same intrinsic parameters of the graphene on Al₂O₃ and graphene on Al₂O₃/metal), they partially interpret this effect in the main text due to "the absence of roughness-mediated scattering and the monocrystalline structure of graphene". Furthermore, the authors introduce more speculations. Although these claims can be fully correct, I think it might be confusing that the increase of the scattering rate is associated to the change of the intrinsic properties of graphene placed above a metal pad. In my opinion, the author's observation is mostly an optical effect, related to different dispersions of the AGPs and GPs. Whether the intrinsic scattering properties of graphene above a metal are affected or not, can be judged by looking into the inverse scattering rate (or lifetime, τ) instead of $\text{Im}(k)/\text{Re}(k)$. The lifetime can be calculated by simply dividing the propagation length, $L=1/\text{Im}(k)$, by the group velocity (the latter can be easily calculated by making a derivation of the dispersion relation, $d\omega/dk$): $\tau=L/v_{gr}$. As the authors can make sure, in theory τ will be the virtually the same for GPs and AGPs. Actually, in the Drude model τ coincides (up to a constant factor) with the inverse scattering rate of the charge carriers. However, if in the experimental measurements (in contrast to the theory) τ of AGP is indeed larger than that of the GPs, then it will be a very clear indication of the change of the intrinsic losses of the charge carriers in graphene.

I encourage the authors to add the discussion on the lifetime and thus clarify a bit the above mentioned paragraph regarding the interpretation of the increased damping rate.

Reviewer #2:

Remarks to the Author:

The manuscript by Menabde et al. reports on their combined experimental and theoretical work on mid-infrared acoustic plasmons in graphene. They use the scattering-type scanning near-field optical microscope to directly image the acoustic plasmons from graphene/Al₂O₃/gold structures and observed Bloch states in devices with embedded gold gratings. The experimental observations agree quite well with their theory and simulation. Overall, the experimental results are interesting and should be useful for the field of graphene plasmonics. I recommend the manuscript in Nature Communications provided that the authors could address the comments below:

(1) On Page 5 of the main text, the authors extracted the Fermi energy $E_f = 0.51$ eV by fitting their data with the FEM simulation of the acoustic graphene plasmon (AGP). On Page 6, they exclude the graphene surface plasmon (GSP) origin of their data by stating that if so, the E_f should be 0.31 eV and this is different than 0.51 eV, inconsistent with the same chemical treatment for the samples. The authors may need to be more careful or provide more evident for this conclusion because the 0.51 eV seems to be based on the AGP assumption (correct me if I am wrong)? Is it possible to attribute their results to GSP and the correct E_f to be 0.31 eV instead? On Page 12, they extracted another $E_f = 0.62$ eV from their devices, and in Fig. 3a, they use $E_f = 0.46$ eV, both are a bit different from 0.51 eV.

Raman or transport data with extracted E_f may provide more support this conclusion.

(2) In Fig. 4, can the author also extract the GSP wavelength from graphene on Al_2O_3 (w/o embedded Au) from similar profile in Fig. 4c or fitting their data with the FEM simulation? Then these data may be added to the dispersion plot in Fig. 3a.

(3) In the controlled experiments in Fig. 6, instead of showing results from two different grating structures with different graphene E_f and at different frequencies, is it less complicated to show data from just one device at various frequencies (of course not at very high frequencies in Fig. S8) for the observation of Bloch modes?

Reviewer #3:

Remarks to the Author:

Real-space imaging of acoustic plasmons in large-area CVD graphene by Menabde et al. performs near-field imaging of acoustic plasmon modes in un-patterned and patterned graphene-dielectric-gold structures. These acoustic graphene plasmons have gained significant interest after recent works showing deep field confinement in the gap region. Among the authors main claims are that: (1) this is the first time near-field imaging is being performed on such excitations (2) that compared to a graphene plasmon on the same dielectric (i.e., no gold), the acoustic plasmon suffers less loss and higher confinement and (3) when patterned, the array does not exhibit coupling resonances when the wavelength of the plasmon matches the array period.

I am not sure as to whether the novelty is sufficient for Nature Communications. On the one hand: acoustic plasmons have been studied many times by now (by far-field imaging, as well as photocurrent nanoscopy). Findings regarding the enhanced confinement are suggested by works such as (Nature nanotechnology 12.1 (2017): 31-35). On the other hand, this is the first near-field imaging study of a timely new platform, as the authors note.

Regardless, I have a few questions and suggestions for improvement regarding the interpretation of the results.

Regarding the confinement and loss in comparison to a "pure" graphene plasmon on a dielectric substrate, the authors attribute the reduced loss to monocrystalline graphene and reduced surface roughness. What is more interesting in my opinion is why an acoustic graphene plasmon might be less lossy than a GSP. Here, the authors suggest that it may arise from screening in the metal improving the intrinsic plasmonic lifetime associated with impurity scattering.

1. Have the authors considered how the electromagnetic energy is distributed within the AGP mode generally? From their Fig. 1, it appears that the AGP mode puts a substantial amount of field inside the spacer layer. If the relative energy inside the spacer vs the graphene sheet is increased, I would expect this to reduce the loss since the mode becomes less sensitive to loss mechanisms associated with the graphene sheet. Moreover, given that the dispersion goes towards higher momenta in the presence of a dielectric surrounding, this would also increase the confinement. It would be nice if the authors could discuss how these considerations play (if they do) into the confinement and loss of the AGP relative to the GSP.

Regarding the patterned array, the authors point out the lack of a resonant coupling in the near-field signal, and attribute it to the behavior of the band structure. I found parts of this discussion confusing. On the one hand, it appears that they attribute a lack of band gaps to loss, but in the supplement, it appears that they consider a case where the losses of the materials are artificially reduced, and there is still no gap (although the bands do look considerably distorted).

2. The authors should further discuss/explain why a resonance was expected in the coupling when the plasmon wavelength matches the array period.

For example, it is not so clear to me that there are any "special frequencies" in the system, as the AGP density of states is similar across the "phase-matching point".

3. Moreover, could the authors comment further on the lack of a bandgap? From an intuitive perspective, one might expect that because the spatial modulation occurs somewhere where there is no field, it can only inefficiently couple backward and forward-propagating plasmons at $k = \pm \pi/P$.

We thank the Reviewers for their valuable comments. We have carefully revised the manuscript in light of all received questions and comments. Our point-by-point responses to them are marked in blue.

=====
Reviewer 1
=====

The paper presents the near-field study (by means of s-SNOM) of the screened (“acoustic”) plasmon polaritons (AGPs) in CVD graphene. This is the first time the AGPs are visualized by purely optical means in CVD graphene. The manuscript is very well written and organized, as is typical for the research teams represented in the study. These results, undoubtedly, will be very interesting and useful for the large part of the community working on the optical properties of 2D materials. I strongly recommend this paper for publication in Nature Communications. However, before the paper is accepted, I would like to make a minor comment.

The authors have experimentally found that the propagation length in terms of the wavelength of the AGPs (the authors call this quantity as the damping rate) is larger than that of the conventional plasmon polaritons in graphene (GPs). Although they find the same increase of the damping rate in the theory (assuming the same intrinsic parameters of the graphene on Al₂O₃ and graphene on Al₂O₃/metal), they partially interpret this effect in the main text due to “the absence of roughness-mediated scattering and the monocrystalline structure of graphene”. Furthermore, the authors introduce more speculations. Although these claims can be fully correct, I think it might be confusing that the increase of the scattering rate is associated to the change of the intrinsic properties of graphene placed above a metal pad. In my opinion, the author’s observation is mostly an optical effect, related to different dispersions of the AGPs and GPs. Whether the intrinsic scattering properties of graphene above a metal are affected or not, can be judged by looking into the inverse scattering rate (or lifetime, τ) instead of $\text{Im}(k)/\text{Re}(k)$. The lifetime can be calculated by simply dividing the propagation length, $L=1/\text{Im}(k)$, by the group velocity (the latter can be easily calculated by making a derivation of the dispersion relation, ω/dk): $\tau=L/v_{gr}$. As the authors can make sure, in theory τ will be the virtually the same for GPs and AGPs. Actually, in the Drude model τ coincides (up to a constant factor) with the inverse scattering rate of the charge carriers. However, if in the experimental measurements (in contrast to the theory) τ of AGP is indeed larger than that of the GPs, then it will be a very clear indication of the change of the intrinsic losses of the charge carriers in graphene.

I encourage the authors to add the discussion on the lifetime and thus clarify a bit the above mentioned paragraph regarding the interpretation of the increased damping rate.

We thank the Reviewer 1 for the valuable feedback.

As noted by the Reviewer, the analytically calculated lifetime $\tau = L/v_{gr}$ of the AGPs and GPs is very similar when graphene model parameters are the same (except for when the alumina loss becomes significant at lower frequencies), as shown below for $E_F = 0.5$ eV and carrier mobility μ of 2000 cm²/Vs:

We agree that the discussion on the intrinsic scattering properties of graphene would require further study for experimental verification; therefore, following the Reviewer’s suggestions, we refrain from discussing this subject in the revised manuscript and focus on the optical effects.

We also realize that the term “damping rate” may cause confusion, even though the term is defined in the manuscript and adopted from previous literatures in graphene plasmonics [Nature 487, 82 (2012)]. To clarify this point, we modified the figure of merit used in the manuscript as the ratio of the plasmon propagation length L_p to its wavelength λ_p : $L_p/\lambda_p = \text{Re}\{k_p\}/(2\pi\text{Im}\{k_p\})$, which is consistent with the definition of FOM as used in [Nanophotonics 9, 2089–2095 (2020); Ref. 8].

Following the Reviewer’s suggestion to clarify points discussed above, **we have revised the figure of merit in the manuscript, and added its detailed analysis into the Supplementary Section S-1, including the discussion on mode field distribution and plasmon lifetime.**

=====
 Reviewer 2
 =====

The manuscript by Menabde et al. reports on their combined experimental and theoretical work on mid-infrared acoustic plasmons in graphene. They use the scattering-type scanning near-field optical microscope to directly image the acoustic plasmons from graphene/Al2O3/gold structures and observed Bloch states in devices with embedded gold gratings. The experimental observations agree quite well with their theory and simulation. Overall, the experimental results are interesting and should be useful for the field of graphene plasmonics. I recommend the manuscript in Nature Communications provided that the authors could address the comments below:

We thank the Reviewer 2 for a kind consideration of our work and valuable comments. We address each of the Reviewer’s questions below.

(1) On Page 5 of the main text, the authors extracted the Fermi energy $E_F = 0.51$ eV by fitting their data with the FEM simulation of the acoustic graphene plasmon (AGP). On Page 6, they exclude the graphene surface plasmon (GSP) origin of their data by stating that if so, the E_F should be 0.31 eV and this is different than 0.51 eV, inconsistent with the same chemical treatment for the samples. The authors may need to be more careful or provide more evidence for this conclusion because the 0.51 eV seems to be based on the AGP assumption (correct me if I am wrong)? Is it possible to attribute their results to GSP and the correct E_F to be 0.31 eV instead? On Page 12, they extracted another $E_F = 0.62$ eV from their devices, and in Fig. 3a, they use $E_F = 0.46$ eV, both are a bit different from 0.51 eV. Raman or transport data with extracted E_F may provide more support this conclusion.

We thank the Reviewer 2 for the opportunity to clarify our argument based on the Fermi levels. The observation of AGP (and not GSP) is supported by the fact that the probed heterostructure with spacer thickness $t \ll \lambda_{AGP}$ does not sustain propagating GSPs due to the proximity of gold to the graphene sheet. For the mid-IR frequencies of interest and $E_F \approx 0.5$ eV, it has been analytically demonstrated in Ref. [8] that the AGP dispersion starts to merge with that of the GSP only when $t \gtrsim 100$ nm. Therefore, considering the sample geometry, only AGP can be excited in the probed samples. **We have modified the discussion in the revised manuscript in order to emphasize this point.**

Regarding the initial argument, as correctly noted by the Reviewer, we estimated the Fermi level in graphene by fitting the experimental data and the plasmon dispersion. Then, we noted the discrepancy between the fitted E_F for samples with 18 and 8 nm spacer thickness: the AGP dispersion fit provided 0.51 and 0.49 eV, respectively, while the GSP fit provided 0.31 and 0.28 eV. The latter values are inconsistent with the cases of chemical doping reported in literature. We agree that E_F by itself cannot be used to pinpoint the type of the observed mode, but it further corroborates our interpretation that the observed mode is the AGP. We also agree that Raman or transport data would also have supported our conclusion, but we did not perform those measurements. **We elaborated on these issues in the revised Supplementary Section S-4.**

(2) In Fig. 4, can the author also extract the GSP wavelength from graphene on Al₂O₃ (w/o embedded Au) from similar profile in Fig. 4c or fitting their data with the FEM simulation? Then these data may be added to the dispersion plot in Fig. 3a.

The data in Fig. 4 was obtained by the near-field imaging of the graphene area near its edge, deposited over the gold nanoribbons embedded in alumina. In this structure, the only sample area supporting the GSP mode is the gap between the gold nanoribbons. However, the gap size is 30 nm, which is much smaller than the GSP wavelength of ~ 300 nm. Therefore, GSP cannot be observed by the s-SNOM anywhere in the structure. The darker area in the images (where near-field contrast is ≈ 1) is the sample surface free of graphene, but gold nanoribbons inside alumina are present everywhere.

As suggested by the Reviewer, we performed the FEM modelling of the near-field signal from both the AGP and the GSP with the device parameters corresponding to Fig. 3. The interference fringes at the graphene edge provide the plasmonic wavelength well matching the analytical dispersion:

The near-field contrast for GSP is much higher due to the absence of gold under the alumina surface, leading to a much weaker signal from the graphene-free area relative to that from the doped graphene sheet. Both plasmons follow the same trend – the near-field signal gets weaker at higher frequency as the plasmon confinement increases. **We have added this additional data into the Supplementary section S-3.**

(3) In the controlled experiments in Fig. 6, instead of showing results from two different grating structures with different graphene E_f and at different frequencies, is it less complicated to show data from just one device at various frequencies (of course not at very high frequencies in Fig. S8) for the observation of Bloch modes?

Indeed, this would be the best case. Unfortunately, due to the depletion of the chemical doping in time, and the fact that the near-field imaging takes considerable amount of time, the results of the measurements vary in time, which limited our ability to collect a wide range of data. In addition, the doping level in graphene, and hence, the exact frequency of the phase matching in the structure, was unknown at the time of the measurements. Therefore, we have a limited number of near-field images at the frequency of phase matching. Figure 6 showcases the best quality data for the two important cases – the AGP phase matching in the array and the slightly detuned array resonance.

We believe that our results spotlight an interesting case of the AGP state in a periodic structure excited at the array resonance. However, we agree that a comprehensive study of AGP Bloch modes requires more experimental investigations and thus is a subject for future investigation. **Consequently, we moved the detailed discussion on the AGP eigenmodes in periodic array of nanoribbons into the Supplementary Section S-6,** which would also improve the flow of the manuscript.

=====
Reviewer 3
=====

Real-space imaging of acoustic plasmons in large-area CVD graphene by Menabde et al. performs near-field imaging of acoustic plasmon modes in un-patterned and patterned graphene-dielectric-gold structures. These acoustic graphene plasmons have gained significant interest after recent works showing deep field confinement in the gap region. Among the authors main claims are that: (1) this is the first time near-field imaging is being performed on such excitations (2) that compared to a graphene plasmon on the same dielectric (i.e., no gold), the acoustic plasmon suffers less loss and higher confinement and (3) when patterned, the array does not exhibit coupling resonances when the wavelength of the plasmon matches the array period.

I am not sure as to whether the novelty is sufficient for Nature Communications. On the one hand: acoustic plasmons have been studied many times by now (by far-field imaging, as well as photocurrent nanoscopy). Findings regarding the enhanced confinement are suggested by works such as (Nature nanotechnology 12.1 (2017): 31-35). On the other hand, this is the first near-field imaging study of a timely new platform, as the authors note.

Regardless, I have a few questions and suggestions for improvement regarding the interpretation of the results.

We appreciate the valuable input given by the Reviewer 3, and thank the Reviewer for their valuable comments. Below we address each of the Reviewer's concerns.

Regarding the confinement and loss in comparison to a "pure" graphene plasmon on a dielectric substrate, the authors attribute the reduced loss to monocrystalline graphene and reduced surface roughness. What is more interesting in my opinion is why an acoustic graphene plasmon might be less lossy than a GSP. Here, the authors suggest that it may arise from screening in the metal improving the intrinsic plasmonic lifetime associated with impurity scattering.

1. Have the authors considered how the electromagnetic energy is distributed within the AGP mode generally? From their Fig. 1, it appears that the AGP mode puts a substantial amount of field inside the spacer layer. If the relative energy inside the spacer vs the graphene sheet is increased, I would expect this to reduce the loss since the mode becomes less sensitive to loss mechanisms associated with the graphene sheet. Moreover, given that the dispersion goes towards higher momenta in the presence of a dielectric surrounding, this would also increase the confinement. It would be nice if the authors could discuss how these considerations play (if they do) into the confinement and loss of the AGP relative to the GSP.

We would like to clarify that for the relative analysis of AGP and GSP modes, we assume the same graphene parameters, and hence the similar plasmon lifetime for both modes. To avoid further confusion, we refrained from discussing the intrinsic loss in graphene in the revised manuscript and **introduced an alternative figure of merit** – the ratio of the plasmon propagation length L_p to its wavelength λ_p : $L_p/\lambda_p =$

$\text{Re}\{k_p\}/(2\pi\text{Im}\{k_p\})$ as in [Nanophotonics 9, 2089–2095 (2020); Ref. 8]; and **focused on the differences between the AGP and GSP field distributions**, as suggested by the Reviewer. **The extended discussion of these points has been added into the revised manuscript and Supplementary section S-1**, as we show below in a detailed manner.

On the one hand, FOM of AGP is generally larger than that of GSP and increases as spacer thickness t decreases, as shown in the figure below (calculated for $E_F = 0.5$ eV and carrier mobility $\mu = 2000$ cm^2/Vs). This effect stems from the AGP dispersion: the rapid compression of AGP wavelength compared to its propagation length, as briefly discussed in [Nanophotonics 9, 2089-2095 (2020); Ref. 8].

On the other hand, we also show that FOM of AGP is indeed less sensitive to the loss in graphene compared to GSP due to the redistribution of the AGP energy flow in the structure. As the spacer thickness t decreases, a portion of the mode's energy flowing inside the dielectric increases, while for the thicker spacer and the GSP mode, a larger fraction of the mode's energy flows along the graphene layer as shown in the left panel below ($E_F = 0.5$ eV, $\mu = 2000$ cm^2/Vs). At the same time, the FOM ratio, $\text{FOM}_{\text{AGP}}/\text{FOM}_{\text{GSP}}$, increases as the ohmic loss in graphene gets larger, which is even more pronounced for thinner spacers (right panel):

Additionally, we have added (into Supplementary Section S-1) a brief discussion on the role of the dielectric medium on the AGP mode confinement relative to the GSP. We show that if the medium above graphene has a larger-than-one dielectric constant, the FOM of GSP and AGP converge due to the symmetrization of the field profiles ($E_F = 0.5$ eV, $\mu = 2000$ cm^2/Vs and $t = 10$ nm for AGP):

Regarding the patterned array, the authors point out the lack of a resonant coupling in the near-field signal, and attribute it to the behavior of the band structure. I found parts of this discussion confusing. On the one hand, it appears that they attribute a lack of band gaps to loss, but in the supplement, it appears that they consider a case where the losses of the materials are artificially reduced, and there is still no gap (although the bands do look considerably distorted).

The absence of bandgap in the structure with negligible material losses stems from the ever-present radiative loss at the nanogap between the gold ribbons. Meanwhile, unless the system is lossless, the Bloch solution of the general dispersion relation for polaritons does not possess a “true” bandgap if solved for the complex wavevector and real frequency [PRL 69, 195111 (2004); Ref. 41]. In order to clarify this point, **we have added an extended discussion** regarding the low-loss and lossless solutions of the Bloch equation into the Supplementary section S-5, as presented below.

In the lossless case, the polaritonic dispersion of the structure can exhibit clearly separated bands and in the band gap region the complex wavevector becomes purely imaginary with its real part being zero. The band gap tends to widen as the reflection at the boundaries increases and the edges of the bands show relatively high density of optical states. As a small amount of losses is introduced to the system, the wavevector in the gap region starts to have a non-zero real component, which looks like a closing of a band gap in the ω vs. $\text{Re}\{k\}$ plot. However, as long as the losses are small, the dispersion still shows a considerable distortion at the band gap region, which leads to a significant variation in the density of optical states. As the losses further increase, the distortion in the dispersion eventually disappears, which can be interpreted as a complete closure of the gap.

There exist two distinctive loss channels in our structure: non-radiative losses due to the ohmic loss in gold and graphene and radiative losses at the boundary (i.e. gap between the gold stripes). In the Supplementary section S-5, we demonstrate the low-loss case by significantly tuning down the non-radiative loss while leaving the radiative loss unchanged. Due to the existence of the radiative loss, the resulting dispersion relation still shows connected bands but the distortion of the band is apparent (both in the real and imaginary wavenumbers) compared to the case with full losses.

Since the radiative loss at the gold gap depends on the geometric shape of the structure, we cannot simply turn it off in rigorous full-wave simulations. Nevertheless, we can still artificially model the truly lossless system by taking the transmission coefficient (δ) from the rigorous simulations and setting the reflection

coefficient (α) in a way that the system satisfies the energy conservation and reciprocity ($|\alpha|^2 + |\delta|^2 = 1$ and $\delta/\delta^* = -\alpha/\alpha^*$). In this case, using the same waveguide parameters as in Fig. S7, the Bloch solution for AGP wavenumber reveals the presence of a band gap. The figure below compares the three cases: full loss, negligible propagation loss, and zero loss:

2. The authors should further discuss/explain why a resonance was expected in the coupling when the plasmon wavelength matches the array period. For example, it is not so clear to me that there are any “special frequencies” in the system, as the AGP density of states is similar across the “phase-matching point”.

Even though the AGP density of states is similar across the measured frequency range, due to the large wavevector mismatch between the free space and the AGP, a far-field excitation of AGP is possible only when the phase matching condition in the array is satisfied. The very efficient far-field coupling to AGP at the phase matching has been demonstrated in Ref. 6 using the similar samples and polarized illumination. In our near-field experiments, the nanoribbons were always orthogonal to the plane of incidence of the TM-polarized illumination beam of the s-SNOM. Therefore, we expected that, in addition to the near-field excitation, the far-field excitation would also contribute to the near-field signal intensity when the phase-matching condition is satisfied. In order to clarify this point, **we have added the pertinent discussion of this into the revised manuscript.**

3. Moreover, could the authors comment further on the lack of a bandgap? From an intuitive perspective, one might expect that because the spatial modulation occurs somewhere where there is no field, it can only inefficiently couple backward and forward-propagating plasmons at $k = \pm \pi/P$.

As we have discussed regarding the bandgap, the combination of high loss (i.e. short propagation length) and weak reflection at the nanogaps prevents a band gap from opening in the polaritonic systems. From the coupling perspective, both of these factors prohibit any significant coupling to the backward and forward propagating modes. One way to overcome this issue could be to reduce the propagation loss (e.g. higher graphene doping and the use of low-loss dielectrics such as CaF_2 or ZnSe) and increase the reflection coefficient at the nanogap.

Reviewers' Comments:

Reviewer #1:

Remarks to the Author:

The authors have carefully addressed the comments from the reviewers, having significantly improved the presentation of the results. In my opinion, the manuscript now can be accepted for publication.

Reviewer #2:

Remarks to the Author:

The authors have addressed my comments, I recommend the publication of the manuscript.

Reviewer #3:

Remarks to the Author:

I have read the response of the authors to all three reviewers, and the corresponding revised text and SM. They have addressed all questions by the reviewers, and I think the work is acceptable for publication in Nature Communications.

We thank the Reviewers for all the valuable comments that helped us to improve the quality of the manuscript. Our responses to them are marked in blue.

=====
Reviewer 1
=====

The authors have carefully addressed the comments from the reviewers, having significantly improved the presentation of the results. In my opinion, the manuscript now can be accepted for publication.

We thank the Reviewer 1 for the valuable feedback and positive evaluation of our work.

=====
Reviewer 2
=====

The authors have addressed my comments, I recommend the publication of the manuscript.

We thank the Reviewer 2 for the valuable comments which contributed to the manuscript revision.

=====
Reviewer 3
=====

I have read the response of the authors to all three reviewers, and the corresponding revised text and SM. They have addressed all questions by the reviewers, and I think the work is acceptable for publication in Nature Communications.

We thank the Reviewer 3 for the valuable comments that contributed to the revision of the manuscript.